# Fragility-Free Global Prescribed Performance Control for Nonlinear Multiagent Systems

1st Shoufeng Yang
*College of Control Science and Engineering*
*Bohai University*
Jinzhou, China
2022008012@qymail.bhu.edu.cn

*Abstract*—In this paper, the fragility-free global prescribed performance control (GPPC) problem for multiagent systems (MASs) is investigated. Firstly, a GPPC strategy is designed for MASs, and the design parameters are independent of the initial conditions, achieving global consensus tracking results. Furthermore, a novel fragility-free control strategy is designed to adjust the performance boundary based on error changes after system stability in real time, avoiding the fragility problem of conventional prescribed performance control schemes. Finally, the proposed control strategy is applied to a simulation example to verify its feasibility and effectiveness.

*Index Terms*—Nonlinear multiagent systems (MASs), fragility-free global prescribed performance control (GPPC).

## I. INTRODUCTION

The analysis and design of multiagent systems (MASs) [1] have been a hot topic in the field of artificial intelligence in recent years. They have broad application prospects in many fields such as aerospace, industrial production, smart grids and parallel computing, and have received widespread attention from researchers. For the needs of large-scale actual engineering, the completion of many control tasks often requires multiple links or combinations to cooperate with each other, and many control systems have complex nonlinearity and high uncertainty. Therefore, more and more scholars are conducting in-depth research on MASs.

The cooperative control of MASs has garnered significant attention due to its widespread applications in various domains, such as robots, unmanned aerial vehicles, smart grids and distributed sensor networks. As a fundamental and important aspect of the cooperative control problem of MASs, the consensus control problem is a research focus of many scholars. A key challenge in this lies in ensuring that the collective behavior of the agents adheres to desired performance specifications while adapting to uncertainties in the environment. To address this challenge, the prescribed performance control (PPC) [2–4] approach has emerged as a promising strategy that enables the design of controllers that explicitly guarantee predetermined transient and steady-state performance. However, the above PPC strategies need to satisfy the initial condition that the initial values of errors are forcibly constrained within the initial performance boundaries, which limits the practicality of the PPC strategies. To solve this problem, Li *et al.* [5] designed a global consensus tracking control strategy for high-order

nonlinear MASs with prescribed performance, which removed the restriction related to the initial condition.

The various uncertainties in complex environments may cause significant disturbances to the systems. If interference causes large amplitudes of errors, they may cause the errors to exceed the performance boundary, which is known as the fragility problem of the PPC approach. Many outstanding results [6, 7] of the fragility-free PPC schemes were proposed, which avoid the challenging fragility problem. Bu *et al.* [6] developed a low-complexity fragility-free PPC approach for constrained waverider vehicles. Most of the existing fragility-free PPC schemes focus on waverider vehicle systems, with few applicable to MASs. As the agent number increases, the control objective changes from ordinary tracking to consensus, which poses certain difficulties in designing fragility-free PPC schemes. This paper aims to design a fragility-free PPC strategy for MASs to avoid the fragility problem.

Taking inspiration from the above, this paper aims to design a fragility-free GPPC scheme for MASs. The main contributions are listed below:

1) A GPPC strategy is designed for MASs, and the design parameters are independent of the initial conditions, achieving global consensus tracking results.
2) A novel fragility-free control strategy is designed to adjust the performance boundary based on error changes after system stability in real time, avoiding the fragility problem of conventional PPC schemes and improving the robustness of the system.

## II. PRELIMINARIES

### A. Graph Theory

Every agent is regarded as a node in MASs. The communication relationships among agents are described by a directed graph $\mathcal{G} = (\mathcal{V}, \mathcal{E}, \mathcal{A})$, where $\mathcal{V} = \{v_1, v_2, \ldots, v_N\}$ is a node set, $\mathcal{E} \subseteq \mathcal{V} \times \mathcal{V} = \{(v_j, v_i)| v_j, v_i \in \mathcal{V}\}$ is an edge set, and $\mathcal{A} = [a_{ij}] \in \mathbb{R}^{N \times N}$ is the adjacency matrix with $a_{ij} > 0 \Leftrightarrow (v_j, v_i) \in \mathcal{E}$ and $a_{ij} = 0 \Leftrightarrow (v_j, v_i) \notin \mathcal{E}$. The digraph $\mathcal{G}$ has no self-loops in this paper. $\mathcal{D} = \text{diag}\{\Sigma_{j=1}^{N} a_{1j}, \ldots, \Sigma_{j=1}^{N} a_{Nj}\}$ is the degree matrix. The Laplacian matrix is described as $\mathcal{L} = \mathcal{D} - \mathcal{A}$. There exists a particular node $v_0$ that can connect each node in the node set $\mathcal{V}$ through directed paths, then the digraph $\mathcal{G}$ contains a spanning tree, where $v_0$ is called to be

the root of the spanning tree. The direct transfer matrix is expressed by the matrix $\mathcal{B} = \text{diag}\{b_1,\ldots,b_N\}$. Moreover, if node $v_0$ can directly transmit information to node $v_i$, $b_i > 0$, otherwise, $b_i = 0$.

*Assumption 1:* [8] The directed graph $\mathcal{G}$ contains a spanning tree, and the leader is the root of the spanning tree.

*Lemma 1:* [8] If a spanning tree is contained in a digraph $\mathcal{G}$, and the leader is its root, then the matrix $(\mathcal{L}+\mathcal{B})$ is nonsingular.

*Lemma 2:* [8] The vectors $\Psi$ and $\Gamma$ satisfy the inequality

$$\Psi^T \Gamma \leq \frac{\grave{\varsigma}\Psi^T\Psi}{2} + \frac{\Gamma^T\Gamma}{2\grave{\varsigma}},$$

where $\grave{\varsigma}$ is a positive constant.

*Lemma 3:* [9] For any continuous function $F(x)$ defined on a compact set $\Lambda_F$, the radial basis function neural networks (RBF NNs) $\bar{\varphi}^T\bar{\Gamma}(x)$ can be used to approximate the function $F(x)$ in arbitrary accuracy $\Xi_F^* > 0$ as follows:

$$F(x) = \bar{\varphi}^T\bar{\Gamma}(x) + \Xi_F(x), \ \forall x \in \Lambda_F,$$

where $\bar{\varphi}$ is the ideal weight vector, $\bar{\Gamma}(x)$ is the basis function vector, and $\Xi_F(x)$ is the approximation error that satisfies $|\Xi_F(x)| \leq \Xi_F^*$.

## III. MAIN RESULTS

### A. Fragility-Free GPPC Strategy and Controller Design

Consider the following nonlinear MAS that contains a leader and $N$ followers. The model of the $i$th ($i = 1, 2, \ldots, N$) follower is expressed as

$$\begin{cases} \dot{x}_{i,k} = x_{i,k+1} + f_{i,k}(\bar{x}_{i,k}), \ k = 1, 2, \ldots, n-1, \\ \dot{x}_{i,n} = u_i + f_{i,n}(\bar{x}_{i,n}), \ i = 1, 2, \ldots, N, \\ y_i = x_{i,1}, \end{cases} \quad (1)$$

where $\bar{x}_{i,k} = [x_{i,1}, x_{i,2}, \ldots, x_{i,k}]^T$ and $\bar{x}_{i,n} = [x_{i,1}, x_{i,2}, \ldots, x_{i,n}]^T$ represent state vectors. $f_{i,k}(\bar{x}_{i,k})$ and $f_{i,n}(\bar{x}_{i,n})$ are unknown nonlinear functions. $u_i$ and $y_i$ are the control input and the output of the $i$th follower, respectively.

The additive mask privacy preservation transformation is constructed as follows:

$$\begin{cases} \xi_{i,k} = x_{i,k} + \delta_{i,k}, \\ \xi_{i,n} = x_{i,n} + \delta_{i,n}, \end{cases} \quad (2)$$

where $\xi_{i,k}$ and $\xi_{i,n}$ are the $i$th agent states under privacy preservation. $\delta_{i,k}$ and $\delta_{i,n}$ are privacy preservation mask functions designed for the $i$th agent.

According to (1) and (2), the $i$th agent under privacy preservation is modeled as follows:

$$\begin{cases} \dot{\xi}_{i,k} = \xi_{i,k+1} - \delta_{i,k+1} + \dot{\delta}_{i,k} + f_{i,k}(\bar{\xi}_{i,k}), \\ \dot{\xi}_{i,n} = u_i + \dot{\delta}_{i,n} + f_{i,n}(\bar{\xi}_{i,n}), \ k = 1, 2, \ldots, n-1, \\ \bar{y}_i = \xi_{i,1}, \end{cases} \quad (3)$$

where $\bar{y}_i$ is the output of the $i$th agent under privacy preservation. $\bar{\xi}_{i,k} = [\xi_{i,1}, \xi_{i,2}, \ldots, \xi_{i,k}]^T$ and $\bar{\xi}_{i,n} = [\xi_{i,1}, \xi_{i,2}, \ldots, \xi_{i,n}]^T$ represent state vectors under privacy preservation.

The distributed consensus tracking error of the $i$th agent is defined as follows:

$$e_{i,1} = \sum_{j=1}^{N} a_{ij}(\xi_{i,1} - \xi_{j,1}) + b_i(\xi_{i,1} - \xi_{0,1}), \quad (4)$$

The prescribed performance error transformation function is designed as

$$\sigma_{i,1} = \frac{\check{m}_i z_{i,1}}{F_{ui} - z_{i,1}} + \frac{\check{m}_i z_{i,1}}{F_{li} + z_{i,1}}, \quad (5)$$

where $\sigma_{i,1}$ is the transformed consensus tracking error. $\check{m}_i$ is a positive constant.

Then, the coordinate transformations of the remaining orders (from 2nd to $n$th order) are given as follows:

$$\sigma_{i,s} = \xi_{i,s} - \alpha_{i,s-1}, \ s = 2, 3, \ldots, n, \quad (6)$$

where $\alpha_{i,s-1}$ is the virtual controller to be designed later.

**Step 1:** The Lyapunov function of the $i$th agent is selected as

$$V_{i,1} = \frac{\sigma_{i,1}^2}{2} + \frac{\tilde{\theta}_i^2}{2\varepsilon_{i,\theta 1}}, \quad (7)$$

where $\tilde{\theta}_i$ represents the adaptive estimation error that satisfies the relationship $\tilde{\theta}_i = \theta_i - \hat{\theta}_i$, where $\theta_i \triangleq \max\{\|\phi_{i,1}\|^2, \|\phi_{i,2}\|^2, \ldots, \|\phi_{i,n}\|^2\}$. $\phi_{i,r}$ ($r = 1, 2, \ldots, n$) is the weight vector of the RBF NNs, and $\hat{\theta}_i$ is the estimated value of $\theta_i$. $\varepsilon_{i,\theta 1}$ is a positive constant.

The derivative of the Lyapunov function $V_{i,1}$ is calculated as

$$\begin{aligned} \dot{V}_{i,1} = &-\check{m}_i z_{i,1}\sigma_{i,1}\left(\frac{\dot{F}_{ui}}{(F_{ui} - z_{i,1})^2} + \frac{\dot{F}_{li}}{(F_{li} + z_{i,1})^2}\right) \\ &+ Q_i\sigma_{i,1}\left(\left(\sum_{j=1}^{N} a_{ij} + b_i\right)\left(\sigma_{i,2} + \alpha_{i,1} - \delta_{i,2} + \dot{\delta}_{i,1}\right.\right. \\ &+ f_{i,1}(\bar{\xi}_{i,1})\right) - \sum_{j=1}^{N} a_{ij}\left(\xi_{j,2} - \delta_{j,2} + \dot{\delta}_{j,1} + f_{j,1}(\bar{\xi}_{j,1})\right) \\ &\left.- b_i\left(\xi_{0,2} - \delta_{0,2} + \dot{\delta}_{0,1}\right)\right) - \frac{\tilde{\theta}_i\dot{\hat{\theta}}_i}{\varepsilon_{i,\theta 1}}, \end{aligned} \quad (8)$$

where $Q_i = \frac{2\check{m}_i\beta_i}{\pi(1+e_{i,1}^2)}\left(\frac{F_{ui}}{(F_{ui}-z_{i,1})^2} + \frac{F_{li}}{(F_{li}+z_{i,1})^2}\right)$.

The virtual controller $\alpha_{i,1}$ of the $i$th agent is designed as

$$\begin{aligned} \alpha_{i,1} = &-\frac{c_{i,1}\sigma_{i,1}}{\left(\sum_{j=1}^{N} a_{ij} + b_i\right)Q_i} + \delta_{i,2} - \dot{\delta}_{i,1} \\ &+ \frac{\check{m}_i z_{i,1}}{\left(\sum_{j=1}^{N} a_{ij} + b_i\right)Q_i}\left(\frac{\dot{F}_{ui}}{(F_{ui} - z_{i,1})^2} + \frac{\dot{F}_{li}}{(F_{li} + z_{i,1})^2}\right) \\ &+ \frac{1}{\sum_{j=1}^{N} a_{ij} + b_i}\left(\sum_{j=1}^{N} a_{ij}\left(-\delta_{j,2} + \dot{\delta}_{j,1}\right) + b_i\left(\xi_{0,2}\right.\right. \\ &\left.\left.- \delta_{0,2} + \dot{\delta}_{0,1}\right)\right) - \frac{\varepsilon_{i,1}}{2}Q_i\sigma_{i,1}\hat{\theta}_i\left\|\zeta_{i,1}(X_{i,1})\right\|^2 - \frac{\varepsilon_{i,2}}{2}Q_i\sigma_{i,1}, \end{aligned} \quad (9)$$

where $c_{i,1}$ is a designed positive constant.

**Step p** $(p = 2, 3, \ldots, n-1)$**:** Choose the Lyapunov function as

$$V_{i,p} = V_{i,p-1} + \frac{\sigma_{i,p}^2}{2}. \quad (10)$$

The derivative of $V_{i,p}$ is calculated as

$$\dot{V}_{i,p} = \dot{V}_{i,p-1} + \sigma_{i,p}\left(\xi_{i,p+1} - \delta_{i,p+1} + \dot{\delta}_{i,p} + f_{i,p}(\overline{\xi}_{i,p}) - \dot{\alpha}_{i,p-1}\right). \quad (11)$$

The virtual controller $\alpha_{i,p}$ of the $i$th agent is designed as

$$\alpha_{i,p} = - c_{i,p}\sigma_{i,p} - \sigma_{i,p-1} + \delta_{i,p+1} - \dot{\delta}_{i,p} + L_{i,p20}$$
$$- \frac{\varepsilon_{i,3p-2} + \varepsilon_{i,3p-1}}{2}\sigma_{i,p} - \frac{\varepsilon_{i,3(p-1)}}{2}\sigma_{i,p}\hat{\theta}_i\|\zeta_{i,p}(\overline{\xi}_{i,p})\|^2, \quad (12)$$

where $c_{i,p}$ is a designed positive constant.

**Step n:** Choose the Lyapunov function as

$$V_{i,n} = V_{i,n-1} + \frac{\sigma_{i,n}^2}{2}. \quad (13)$$

The derivative of $V_{i,n}$ is calculated as

$$\dot{V}_{i,n} = \dot{V}_{i,n-1} + \sigma_{i,n}\left(u_i + \dot{\delta}_{i,n} + f_{i,n}(\overline{\xi}_{i,n}) - \dot{\alpha}_{i,n-1}\right). \quad (14)$$

The controller $u_i$ of the $i$th agent is designed as

$$u_i = - c_{i,n}\sigma_{i,n} - \sigma_{i,n-1} - \dot{\delta}_{i,n} + L_{i,n20}$$
$$- \frac{\varepsilon_{i,3n-2} + \varepsilon_{i,3n-1}}{2}\sigma_{i,n} - \frac{\varepsilon_{i,3(n-1)}}{2}\sigma_{i,n}\hat{\theta}_i\|\zeta_{i,n}(\overline{\xi}_{i,n})\|^2, \quad (15)$$

where $c_{i,n}$ is a designed positive constant.

### B. Stability Analysis

*Theorem 1:* Consider the MAS (3). Under *Assumption 1*, the virtual controllers (9), (12), the controller (15), the following control objectives can be guaranteed that

- All signals of the MAS are bounded.
- The consensus tracking errors of all agents can achieve the predetermined accuracy.

*Proof:* The total Lyapunov function of $N$ agents can be selected as

$$V_N = \sum_{i=1}^{N} V_{i,n}$$
$$= \sum_{i=1}^{N} \left( \sum_{s=1}^{n} \frac{\sigma_{i,s}^2}{2} + \frac{\tilde{\theta}_i^2}{2\varepsilon_{i,\theta 1}} \right). \quad (16)$$

The derivative of $V_N$ is calculated as

$$\dot{V}_N = \sum_{i=1}^{N} \dot{V}_{i,n}$$
$$\leq \sum_{i=1}^{N} \left( -\sum_{s=1}^{n} c_{i,s}\sigma_{i,s}^2 - \frac{c_{i,\theta}}{2\varepsilon_{i,\theta 1}}\tilde{\theta}_i^2 + \Delta_{i,n} \right)$$
$$\leq -\kappa V_N + \gamma. \quad (17)$$

where $\kappa = \min\{2c_{i,1}, 2c_{i,2}, \ldots, 2c_{i,n}, c_{i,\theta}\}$, $\gamma = \sum_{i=1}^{N} \Delta_{i,n}$.

## IV. SIMULATION RESULTS

The trajectory of the leader vehicle is given as $y_0 = -2\sin(0.1\pi t)$. Based on [10], the $i$th $(i = 1, 2, \ldots, 6)$ follower vehicle is modeled as

$$m_i\ddot{y}_i = \upsilon_i - \mu m_i g - \tau_i \dot{y}_i, \quad (18)$$

where $y_i$ and $\upsilon_i$ are the output and the control force of the $i$th vehicle, respectively. $m_i$ is the total mass of the $i$th vehicle. $\mu$ is the rolling friction coefficient. $g$ is the gravitational acceleration. $\tau_i$ is the unknown viscous friction coefficient of the $i$th vehicle.

Select the state variables $x_{i,1} = y_i$, $x_{i,2} = \dot{y}_i$, the relationship (18) can be expressed as the following double integrator model

$$\begin{cases} \dot{x}_{i,1} = x_{i,2}, \ i = 1, 2, \ldots, 6, \\ \dot{x}_{i,2} = u_i - \mu g - \dfrac{\tau_i}{m_i}x_{i,2}, \\ y_i = x_{i,1}, \end{cases} \quad (19)$$

where $\mu = 0.015$, $g = 9.8$ m/s$^2$, $\tau_1 = \tau_2 = \tau_5 = 0.2$ kg/s, $\tau_3 = \tau_4 = \tau_6 = 0.4$ kg/s and $m_i = 2$ kg.

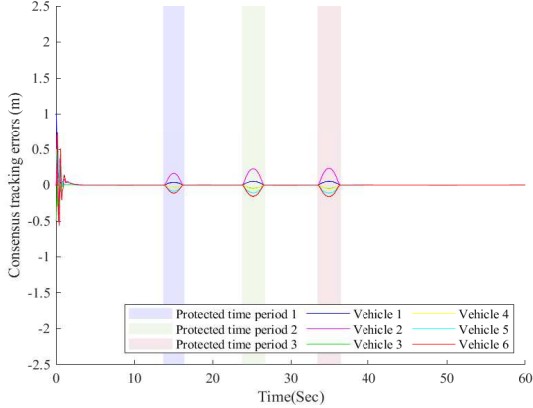

Fig. 1. The tracking trajectories of the multi-vehicle system.

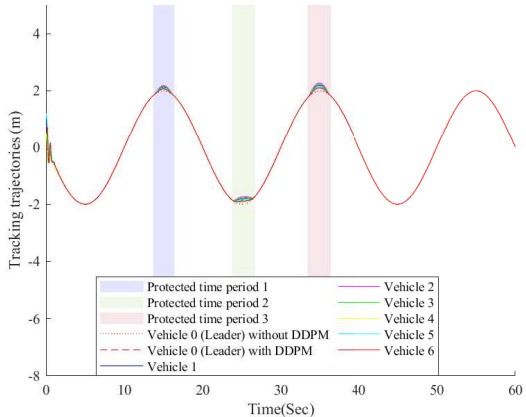

Fig. 2. The consensus tracking errors of the six vehicles.

The evolution of the tracking trajectories for the multi-vehicle system is shown in Fig. 1. We can see that the trajectories deviate during some time periods, which are exactly the time periods. We can see that the trajectories deviate during some time periods, which are precisely the time periods that have been strongly disturbed.

Fig. 2 shows the consensus tracking errors of the six vehicles. During the three protected time periods, the consensus tracking errors of the six vehicles increase rapidly due to the action of strong disturbance but quickly converge to near zero.

## V. Conclusion

In this paper, the fragility-free GPPC problem for MASs has been investigated. Firstly, a GPPC strategy has been designed for MASs, and the design parameters are independent of the initial conditions, achieving global consensus tracking results. Furthermore, a novel fragility-free control strategy has been designed to adjust the performance boundary based on error changes after system stability in real time, avoiding the fragility problem of conventional prescribed performance control schemes. Finally, the proposed control strategy has been applied to a simulation example to verify its feasibility and effectiveness.

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
