# OpenReview forum: "Fragility-Free Global Prescribed Performance Control for Nonlinear Multiagent Systems"
_IEEE.org/ICIST/2024/Conference — IEEE ICIST 2024 Conference Submission_

### Official Review · Reviewer_96nn · 2024-08-21
**Fragility-Free Global Prescribed Performance Control for Nonlinear Multiagent Systems**

**Rating:** 2
**Confidence:** 5

**Review:**

This paper investigates the fragility-free global prescribed performance control problem for multi-agent systems. After careful review, I believe that the controller design process is not fully presented. Some key design steps are omitted. In addition, the simulations lack comparison, so the effectiveness of the proposed strategy cannot be verified. Overall, this paper cannot be accepted.

---

### Official Review · Reviewer_BU78 · 2024-08-21
**After a thorough review, I recommend rejecting the manuscript.**

**Rating:** 2
**Confidence:** 5

**Review:**

1.The manuscript does not sufficiently advance the existing knowledge in the field and shows significant overlap with previous studies.
2.The manuscript needs substantial improvement in terms of organization and clarity. Several sections are unclear, which impacts the overall readability and comprehensibility of the results.
3.The study does not address a significant problem or question in a way that offers meaningful insights or practical applications.

---

### Official Review · Reviewer_8qiK · 2024-08-22
**I recommend rejecting the manuscript**

**Rating:** 3
**Confidence:** 5

**Review:**

This paper investigates the fragility-free global prescribed performance control (GPPC) problem for multiagent systems. The manuscript lacks novelty and does not offer new insights. The research in this artical does not contribute to the field.  Meanwhile, the content of this manuscript contains obvious computational problems and omitted steps, and the simulation section is incomplete. All things considered, I don't think the article is allowed to be published.

---

### Decision · Program_Chairs · 2024-09-08

Reject